# Nucleic Acid Vaccine Platform for DENGUE and ZIKA Flaviviruses

**DOI:** 10.3390/vaccines10060834

**Published:** 2022-05-24

**Authors:** Jarin Taslem Mourosi, Ayobami Awe, Swati Jain, Himanshu Batra

**Affiliations:** 1Department of Biology, The Catholic University of America, Washington, DC 20064, USA; taslemmourosi@cua.edu (J.T.M.); awe@cua.edu (A.A.); 2Department of Surgery (Head and Neck Service), Memorial Sloan Kettering Cancer Center, New York, NY 10065, USA; 3Program in Cellular and Molecular Medicine, Boston Children’s Hospital, Harvard Medical School, Boston, MA 02115, USA

**Keywords:** mRNA vaccine, DNA vaccine, Dengue and Zika

## Abstract

Dengue virus and Zika virus are mosquito-borne, single-stranded, positive-sense RNA viruses that belong to the Flaviviridae family. Both the viruses are closely related and have similarities with other flaviviruses. Dengue virus (DENV) causes a severe febrile illness with fever, joint pain, and rash leading to a life-threatening condition in severe cases. While Zika virus (ZIKV) primarily causes mild fever, it can be passed from a pregnant mother to her fetus, resulting in severe birth defect microcephaly and even causing a rare autoimmune disease—Guillain–Barre syndrome. To date, there are no approved DENV and ZIKA vaccines available, except a Dengue vaccine (Dengvaxia, Sanofi Pasteur Inc., Lyon, France) recently approved to be used only for 9–16 years of age groups living in endemic areas and having a previous record of confirmed dengue infection. There are several potential vaccine candidates in the clinical trials based on multiple vaccine platforms, such as live attenuated, subunit, nucleic acid, and viral vector-based vaccines. In the current review, we have focused exclusively on the nucleic acid vaccine platform and discussed the progress of all the DNA/RNA vaccine candidates under preclinical and clinical studies for DENV and ZIKA viruses. Additionally, we have described a brief history of the emergence of these flaviviruses, major structural similarities between them, prominent vaccine targets, and the mechanism of virus entry and infection.

## 1. Introduction

Both Dengue virus (DENV) and Zika virus (ZIKV) are enveloped icosahedral flaviviruses belonging to the Flaviviridae family. The transmission for both viruses is through bites of Aedes mosquitos, such as *A. aegypti* and *A. albopictus*. Historically, dengue fever dates to 1779–1780, with the first reported epidemics in Asia, Africa, and North America [1]. Recently, the global incidence of dengue has been estimated to be 100–400 million per year [2]. A study revealed that more than 3.96 billion people among 128 countries are at risk of dengue virus infection [3]. In 1947, ZIKV was first documented in monkeys in Zika forest, Uganda [4]. Then, in 1960, the first human case was reported in Nigeria [5]. The virus circulation was self-limiting until the Yap Island (2007) and Micronesia epidemics [6]. However, later on several outbreaks occurred between 2013–2016 in French Polynesia [7], northern Brazil [8,9] central South America [10,11], Puerto Rico [12], and the U.S. Virgin Islands [13].

Dengue and Zika infection can be either asymptomatic or symptomatic. Dengue infection can be further classified into dengue fever (DF) and dengue hemorrhagic fever (DHF). The major symptoms of DF are febrile illness, headache, joint pain, rash, and muscle pain that can last for 1–2 weeks. In severe dengue or DHF, plasma capillarity leakage, thrombopenia, and mild to severe organ damage can occur, leading to life-threatening conditions [14]. The severity of dengue virus infection depends on virus serotype, patient age, sex, genetic and immunological condition, and, importantly, prior infection with dengue virus or other related viruses. Dengue virus has four antigenically distinct serotypes (DENV1, DENV2, DENV3, DENV4) that share 65–70% nucleic acid identity [15]. Infection by one of the serotypes elicits lifelong immunity to the serotype, but immunity to other serotypes is temporary, lasting less than six months. Secondary infection by different serotypes can leads to deadly dengue symptoms due to antibody-dependent enhancement (ADE) [16]. Like DENV, ZIKV infection is mostly asymptomatic; only 20% of the affected individuals show mild fever-like symptoms. However, ZIKV can be transmitted through an infected mother to fetus and male to female via semen, causing microcephaly, a severe congenital disability characterized by the reduced head size of the baby that leads to developmental delay and other serious complications [17]. In adults, ZIKV infection is linked to a rare autoimmune disease called Guillain–Barre syndrome [18], in which the immune system of the affected individual attacks the peripheral nerves, resulting in paralysis.

Currently, there is no approved vaccine for Zika, whereas dengue has only one approved vaccine, Dengvaxia, which is a tetravalent live attenuated vaccine made from a yellow fever 17D viral strain. This vaccine construct was made by replacing two proteins of yellow fever virus (precursor membrane and envelope) with dengue counterparts. This vaccine is not effective against all four dengue strains. It shows only 34.7% efficacy against DENV2, which causes the most severe infection. Moreover, this vaccine increases the risk of severe infection in seronegative vaccinated individuals. Therefore, this vaccine is only approved for young individuals aged 9 to 16 with previous laboratory confirmed dengue infection. Currently, there are several vaccine candidates for DENV and ZIKV in the different preclinical and clinical trial stage. In this review, we describe the structural aspects of both dengue and Zika viruses, and also review the recent advances in the development of nucleic acid vaccines to prevent their infections.

## 2. Dengue and Zika Virus Structure

DENV and ZIKV have single-stranded and positive-strand RNA genomes (~10.8 kilobase RNA), translated to a single polyprotein (Figure 1). Upon translation, a large polyprotein is produced having a 5′ and 3′ untranslated region. After the cleavage, polyprotein produces three structural proteins: capsid protein (C), precursor membrane protein (prM) and envelope protein (E), and seven non-structural proteins (NS1, NS2A, NS2B, NS3, NS4A, NS4B, NS5) [19].

Capsid protein protects the viral RNA, while envelope protein mainly interacts with the host cell receptor, and precursor membrane (prM) protects E protein from premature fusion to endosome. prM is cleaved into pre-membrane and membrane. Non-structural proteins have a role in virus replication, genome packaging, and escaping from the host immune system. NS1 helps in the replication process, NS3 has helicase and protease activity, and NS5 is a large non-structural protein that acts as a polymerase and methyltransferase.

Among three structural proteins, E protein is present outside the surface (Figure 1a), whereas M protein is found predominantly inside, both positioned parallelly. There is a total of 180 copies of E and M proteins. E protein dimer follows two-fold symmetry and has four beta-stranded domains; DI, DII, DIII encodes the viral envelope’s surface portion and stem transmembrane domain and anchors protein to the lipid membrane (Figure 2a). DI and DII domains carry the highly hydrophobic fusion loop required to initiate the infection, and DIII has the putative receptor-biding site, which interacts with the host cell membrane.

Both DENV and ZIKV E proteins are conserved with other flaviviruses. E protein contains a glycosylation site that plays a crucial role in virus infectivity and tropism [20]. The E protein is attached to the host receptor during viral entry, and the N-linked glycosylation creates a binding site receptor attachment. E protein of ZIKV has one glycosylation site at (Asn154) while DENV has two sites (Asn67, Asn153) [21]. The capsid protein of ZIKV forms a dimer having a two-fold symmetry, and each monomer has four alpha-helices connecting via a loop.

## 3. Virus Entry and Infection

Both DENV and ZIKA viruses can enter the host cell through various receptors. DC-SIGN, GRP-78, laminin receptor, and heat shock protein 70 (HSP 70) are potential receptors for dengue virus [22,23,24,25], while DC-SIGN, Tyro3, Phosphatidylserine receptors, and heparan sulfate are the putative receptors for ZIKV that can initiate Clathrin-mediated viral entry (Figure 3) [26,27]. Alternatively, the virus can also bind to sub-neutralizing antibodies that antibody present due to past infection. Upon entry into the cell, the acidic environment of the endosome mediates the fusion of the virus with the membrane, and the viral RNA is released into the cytoplasm. The viral genome is translated into ten proteins and processed in the endoplasmic reticulum (ER) lumen. During viral assembly at ER, a non-infectious virus particle is formed, which has 60 copies of trimeric E-prM spike, then the virus moves through the trans-golgi network. The lower pH in golgi activates furin protease that cleaves prM to expose the E protein fusion loop. Then, the virus can fuse with the host membrane, releasing itself from the host cell.

## 4. Nucleic Acid Vaccine Platform

Current vaccine approaches that are being used for the development of DENV and ZIKV include live attenuated and inactivated constructs, recombinant antigens, DNA constructs, vector-based expression, and virus-like particles (VLPs) [29,30]. Herein, we have focused on the developmental progress and the immune responses of only nucleic acid-based vaccines, specifically DNA and mRNA-based vaccines for these flaviviruses.

The first reported use of nucleic acid construct was in 1990, when a study showed that injection of DNA and RNA expression vectors into mouse skeletal muscle results in protein expression [31]. Basically, in a nucleic acid-based vaccine platform, the genetic material, DNA or RNA encoding a specific protein on the target pathogen, is inserted into host cells. Upon insertion, the host’s transcriptional and translational machinery is used to produce the desired gene product, also known as antigen, which then triggers an immune response [32].

In DNA vaccines, an antigenic gene or gene of interest is inserted in plasmid DNA, and it is delivered intradermally (i.d) or intramuscularly (i.m). DNA vaccines encode antigens, which can be expressed in both MHC class I and class II pathways; it can activate CD8+ and CD4+ T cell and antibody responses (Figure 4). When a DNA vaccine is injected into the cell i.d or i.m, cells uptake the DNA, plasmid DNA is processed and elicit immune response using three different mechanisms: (i) direct priming by somatic cell, (ii) direct transfection of dendritic cell or antigen-presenting cell (APC), and (iii) cross-priming [33]. A DNA vaccine is stable, shows higher safety, long-term expression of immunogens, and is rapidly produced [34]. DNA vaccines are easier to manufacture than live attenuated or subunit vaccines, as it does not require complicated protein purification process [35].

mRNA has recently become a focal point in vaccine development due to its advantages over conventional vaccine platforms (such as live attenuated and inactivated pathogens and subunit vaccines). The safety, efficacy, and ease of production of mRNA makes it an ideal candidate for vaccine production [36]. The efficacy of mRNA vaccines against viral pathogens and the use of mRNA vaccines in combating different infectious diseases, such as COVID-19, HIV, influenza, rabies, and chikungunya, have been reported in various studies [37,38,39,40,41].

For mRNA vaccines, the mode of delivery is a critical aspect of development as the exogenous mRNA needs to cross the lipid bilayer membrane into the cytoplasm, where it is translated into a functional protein. Barriers to the intracellular delivery of naked mRNA into the cytoplasm include its large molecular weight, high negative charge, instability, and susceptibility to nuclease digestion. Hence the need for an efficient delivery system to overcome this barrier. Different approaches have been explored for the efficient delivery of mRNA into cells. These include ex vivo loading of mRNA into dendritic cells (DCs) via endocytic pathways or electroporation; the use of physical methods, such as a gene gun to penetrate the lipid bilayer of the cell; complexing of protamine to the mRNA to protect the mRNA from degradation; and the use of cationic lipid and polymer-based delivery, such as lipofectamine, dendrimers and lipid nanoparticles (LNPs). Of the aforementioned delivery systems, LNPs have become the most commonly used delivery system, owing to their ability to self-assemble, mimic the lipid bilayer of the cell, and allow the endosomal release of its content into the cytosol after cellular uptake via endocytosis (Figure 4) [42,43,44].

**Figure 4 vaccines-10-00834-f004:**
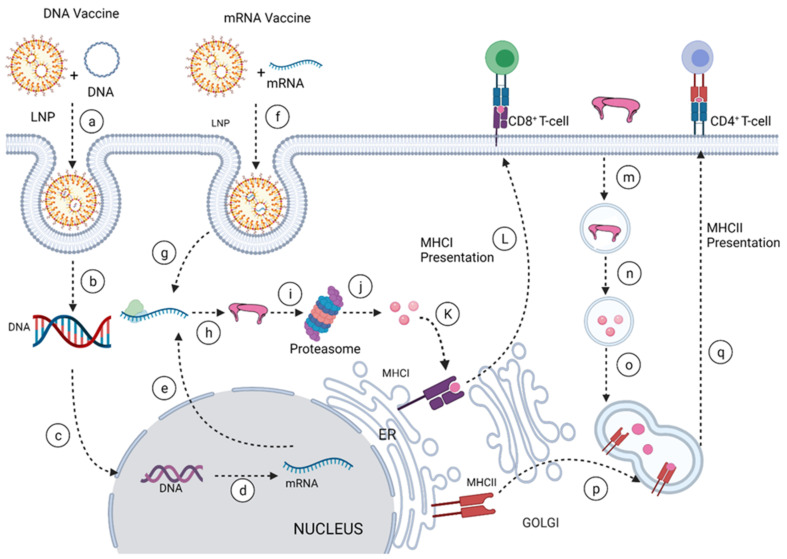
Nucleic acid vaccine delivery method [43]. The LNP formulated DNA or mRNA is taken up into the cell via endocytosis (a, f). Once inside the cytosol, in a process referred to as endosomal release, the naked DNA or mRNA is released into the cytosol (b, g). The DNA is transported into the nucleus (c), where it is transcribed into mRNA (d), and transported out of the nucleus into the cytosol (e). The mRNA is then translated into protein (h), which can either be released outside the cell via the Golgi, a process called exocytosis, or it can be taken up and degraded into peptides by the proteasome (i, j). The peptide is displayed on MHC class I (MHC I) on the ER (k), the MHC I together with the displayed peptide is ferried via Golgi to the cell membrane (l), where the peptide is presented to the CD8+ T cell, thus leading to cell-mediated immunity. The released protein can be taken into the cell (m) via endocytosis and degraded into smaller peptides in the vesicle compartment (n). MHC class II (MHCII) is processed from the ER to the Golgi in another vesicular compartment which is fused with the vesicle containing the peptides (o, p), wherein the peptide is displayed on MHCII and ferried to the cell membrane where the peptide is presented to the CD4+ T cell (q), a downstream process that leads to the activation of antibody-mediated immunity.

## 5. Progress of the Nucleic Acid Vaccines for DENV and ZIKV and Their Responses

### 5.1. DNA Vaccines for DENV

A monovalent DNA vaccine for DENV2 was constructed by cloning of the prM gene and 92% envelope gene from NGC DENV-2 strain in eukaryotic plasmid vectors, and the carboxy terminus of E protein was truncated to improve the secretion of the E protein (Figure 5a) (45). BALB/c mice immunized with this vaccine elicited anti-dengue antibodies that were able to neutralize the DENV2 strain in vitro [45]. Another study showed that the presence of the CpG motif in the plasmid improves the antibody response against the DENV2 strain, and this vaccine showed a protective effect in the mouse challenge model (Figure 5b) [46]. Later, Jimenez et al. showed a truncated E protein for DENV2 was unable to elicit complete protection in mice due to weak immune system activation, which relates to E protein improper secretion and absence of prM protein [47]. To optimize the DNA vaccine delivery and proper secretion, a lysosome-associated membrane protein (LAMP) was incorporated into the envelope protein of DENV2; this modified DNA vaccine could effectively target MHC-class II and showed a more robust neutralizing antibody response in immunized mice [48]. Blair et al. designed a recombinant DNA vaccine incorporating E protein and prM of DENV3 that showed moderate DENV-specific antibody response and imparted partial protection against the virus challenge [49].

A phase-I clinical trial was performed for a prototype monovalent DNA vaccine against DENV1 (D1ME), incorporating prM and E gene, the vaccine proved safe for human use; however, it was unable to provide a robust neutralizing antibody response and was found to be less immunogenic [50]. To improve the humoral response of the DNA vaccine, an adjuvanted plasmid Vaxfectin was used, which is the combination of cationic and neutral lipids; this adjuvanted vaccine showed an increased anti-dengue response as well as protection in the DENV2 challenge study [51]. Using the adjuvant, a prM and E-based tetra-valent DNA vaccine was constructed and tested for phase-I clinical trial without adjuvant, and with adjuvant, the anti-dengue antibody response improved significantly after adding the adjuvant. However, the T cell response remained unchanged compared to the non-adjuvanted vaccine [52].

The prM and E protein induce neutralizing antibodies to DENV [53]. T cell immune response from structural protein (prM and E) [54,55] and non-structural protein 3 (NS3) also elicit a cellular immune response [56,57]. Non-structural protein 1 (NS1) can alone induce a protective immune response as shown by Costa et al. They constructed NS1 and NS3 based DNA vaccines for DENV2, of which the former candidate induced a high level of NS1 specific antibody response in the BALB/c mice [58] and the latter resulted in protection via induction of interferon-γ [59].

A DNA vaccine using prM and E protein of DENV1-4 showed protection in mice from lethal dengue challenge [60,61,62,63]. A vaccine against DENV3 also showed cross-protection against the DENV1, DENV2, DENV4, which is important for tetravalent dengue vaccine design [63]. Multiple immunizations using electroporation improved the immunogenicity significantly and showed robust antigen-specific T cell response and persistent antibody response [60,61,62].

### 5.2. DNA Vaccines for ZIKV

Currently, there is no approved vaccine available for ZIKV. To date, 13 different vaccines have completed the phase I trial, and only one DNA vaccine has finished the phase II trial (Table 1). Larocca et al. developed several DNA vaccine constructs for Zika based on prM, E protein and their mutated versions (Figure 5e) [64]. They removed 93 amino acids from prM in one of the constructs, and also deleted transmembrane (ΔTM) and stem region (Δstem) of the envelope protein separately and cloned these constructs in a eukaryotic expression plasmid (pcDNA3.1+) vector with Kozak sequence and Japanese encephalitis virus leader sequence [64]. When 50μg of this DNA vaccine was injected in BALB/c mice via the intramuscular route, higher Env specific antibody response was elicited in the prM-Env vaccine than the other five constructs (Figure 5e [2,3,4,5,6]). Even a single dose of the prM-Env vaccine could elicit neutralizing antibody response and show protection against the ZIKV challenge [65].

Dowd et al. constructed two DNA vaccines encoding full length of prM and envelope of French Polynesian ZIKV isolate, where prM signal sequence of ZIKV was exchanged with Japanese encephalitis virus (JEV) signal sequence which improved the expression level (Figure 5f) [35]. To improve the subviral particle (SVP) formation, one construct (VRC 5288) was further modified by incorporating stem and transmembrane from JEV. Single immunization of BALB/c and C57BL/6 mice with 50 μg of the DNA vaccine elicited ZIKV specific neutralizing antibodies with EC50 titer of 10^5^ in BALB/c mice [35]. Both the vaccine candidates were immunogenic in mice and non-human primate models and competed in phase I clinical trial.

VRC5288 completed its phase I clinical trial in 2019 (Clinical trial NCT02840487, study VRC319). In a total of 80 participants (in four groups), 4 mg single doses of vaccine were administered via intramuscular injection in the deltoid muscle, with boosts at week 4, week 12, week 4 and 8, and week 4 and 20 where higher seroconversion 89% was reported on week 4, and week 12 boost with neutralizing antibody GMT titer of 120 (73–197).

VRC5283 (Clinical trial NCT02996461, study VRC 320) elicited neutralizing titer GMTs 304 (215–430) when a 4 mg single dose was posed at week 4 and 8. Further, 100% neutralizing antibody seroconversion was observed. VRC 5283 has also completed the phase II clinical trial.

Another ZIKV DNA vaccine (GLS-5700) was constructed using consensus prM and envelope gene sequence (ZIKV strain isolated from 1952 to 2015), utilizing the modified pVax1 expression vector where immunoglobulin E (IgE) leader sequence was added to improve the expression level (Figure 5g) [66]. In phase I clinical trial of this vaccine (Clinical trial NCT02809443), 1 mg and 2 mg doses of the DNA vaccine were administered into two groups (20 participants each), ZIKV specific neutralizing antibody development was reported in 60% and 63% of the participants of each group, respectively, with MN50 titer in the range of 1:18 to 1:317 [67]. Furthermore, intraperitoneal administration of 0.1 mL post-vaccination serum to interferon (α and β) receptor (IFNAR) knockout mice was able to protect 92% of the mice challenged with lethal ZIKV. This vaccine was delivered via electroporation using a CELLECTRA-3P device which showed a better immune response than a simple injection [67].

Although above mentioned, a DNA vaccine for ZIKV is based on prM and envelope proteins that confer only neutralizing antibody mediated protection. However, cellular immunity is equally critical to clear ZIKV infection. NS1 is an attractive target for induction of cellular immune response as it can alone stimulate T cell mediated immune response against ZIKV [65].

### 5.3. mRNA Vaccines for DENV

In 2019, Roth et al. reported that the prime-boost immunization of humanized HLA class 1 transgenic mice with DENV1-NS, a vaccine that comprised of the most immunogenic portions of NS3, NS4B, and NS5, induced a strong CD8+ T cell immune response and conferred significant protection after challenge with DENV1 (Figure 6d) [68].

Moreover, using mRNA encapsulated by lipid nanoparticles (LNPs), Zhang et al. developed a DENV mRNA vaccine based on two structural proteins—prME and E80 (containing 80% N-terminal of the ENV protein)—and one non-structural protein (NS1) from DENV2 (Figure 6c) [69]. All mRNA elicited virion-binding antibodies; however, E80-mRNA yielded the strongest neutralizing antibody response against DENV2 with PRNT50 of 11,000 in BALB/c mice. Furthermore, vaccination with the combination of E80+NS1 mRNAs elicited antigen-specific T cell responses [69].

Wollner et al., demonstrated in immunocompromised AG129 mice that a nucleotide modified mRNA vaccine encodes DENV1 prM and E protein under JEV signal peptide elicited neutralizing antibodies, with EC50 titer of 1/400 ]. Anti-DENV1 CD4+ and CD8+ T cells were also activated. They also designed some E protein mutants, G106R, L107D, and F108A, to remove the fusion loop, and both the vaccines reduced ADE of DENV2 in K562 cells (Figure 6a,b) [70].

### 5.4. mRNA Vaccines for ZIKV

Two modified lipid nanoparticle packaged mRNA vaccines have been constructed (Figure 6) [71]. The first one contains (Figure 6e) a human IgE signal sequence with the full length of prM and E gene from an Asian strain (Micronesia 2007), and booster doses of this construct produced potent serum neutralizing titer and neutralizing immunity. In the second construct design, four mutations were inserted in the E protein T76R, Q77E, W101R, and L107R under JEV’s signal sequence to prevent cross-reacting antibodies (Figure 6f) [71].

In 2017, a study demonstrated that immunization with mRNA-LNP encoding the pre-membrane and envelope (prM-E) glycoproteins of ZIKV confers protection in mice and non-human primates. In this study, immunization of C57BL/6 mice with a 30 μg single dose of nucleoside-modified ZIKV prM-E mRNA-LNP vaccine-elicited E protein-specific CD4+ T cell responses. Further, vaccinated BALB/c mice and rhesus macaques (four out of five) showed no detectable viremia at day seven post ZIKV challenge [36].

Chahal et al. developed a modified dendrimer nanoparticle (MNDP) based RNA replicon vaccine [72]. This vaccine incorporated prM and E gene from an Asian lineage ZIKV isolate Z1106033, and the vaccine was able to elicit an E protein-specific IgG response in C57BL/6 mice. This study also identified an MHC I restricted 9-mer epitope IGVSNRDFV, which produces CD8+ T cell response in C57BL/6 mice (72).

Another self-replicating RNA vaccine utilizes an attenuated strain of the Venezuelan equine encephalitis (VEE) virus, TC-83. This vaccine expresses prM and E genes of ZIKV with ZIKV and Japanese encephalitis virus (JEV)’s signal sequence under a T7 promoter in the VEE replication vector, delivered through a stable nanostructured lipid carrier. A 10-ng single dose of this vaccine showed protection in mice against the lethal ZIKV challenge (Figure 6g) [73].

In another study, using a prime-boost strategy, intradermal electroporation of 1 μg of a naked self-replicating mRNA ZIKV vaccine encoding the ZIKV prM-E elicited high antibody titers in IFNAR1 knockout C57BL/6 mice, and all the mice were protected against the ZIKV challenge. The vaccine also conferred complete protection in vaccinated mice, while a four out of seven mortality was observed in the control group [74].

Luisi et al. developed another self-amplifying RNA (SA-RNA) vaccine encoding capsid, prM, and Env. A total of 1.5 μg of this SA-RNA vaccine delivered via cationic nanoemulsion was enough to produce neutralizing antibody responses in BALB/c mice [75].

## 6. Concluding Remarks

DENV and ZIKV would continue to cause serious global health concerns in the absence of effective vaccines. However, several nucleic acid vaccines for DENV and ZIKV have been tested in phase I clinical trials that have shown promising results, with one of the mRNA vaccines for ZIKV in phase II clinical trial. Most of the vaccine constructs are focused on either structural proteins (prM or envelope), or non-structural proteins (NS1 or NS3), which are unable to effectively elicit both neutralizing antibody and cellular immune responses. Therefore, an ideal vaccine construct might include both structural and non-structural antigens, excluding any target epitope with a potential of inducing cross-reactive antibody responses. Since a nucleic acid vaccine is relatively less immunogenic, the vaccine target should be good enough to induce robust immune response. In light of the ongoing COVID-19 pandemic, nucleic acid-based vaccines have turned out to be an attractive platform for rapid generation of the vaccine and it certainly has the potential to become a next-generation vaccine production platform for other viruses as well.

## Figures and Tables

**Figure 1 vaccines-10-00834-f001:**
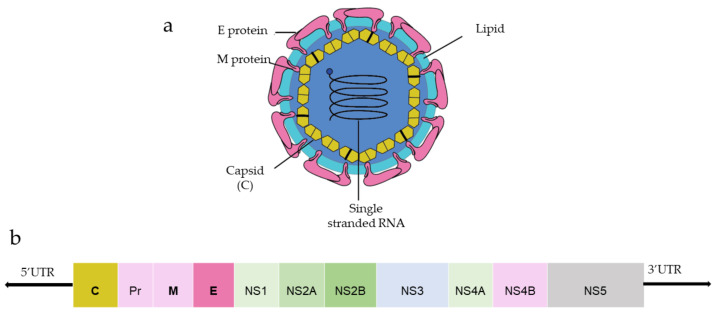
(**a**) General structure of the flavivirus; (**b**) genomic organization of the flavivirus (C, prM, E are structural proteins; NS1, NS2A, NS2B, NS3, NS4A, NS4B, NS5 are non-structural proteins).

**Figure 2 vaccines-10-00834-f002:**
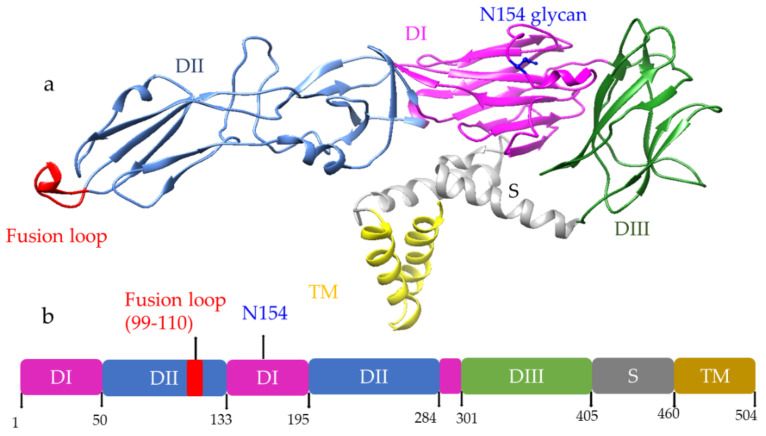
(**a**) Structure of E protein monomer (PDB ID: 5IRE) of ZIKV is shown in ribbon form. (**b**) Schematic of E protein showing residue positions of the all the domains. The domains are color coded as shown. Domain I (DI); Domain II (DII); Domain III (DIII); Stem (S); Transmembrane (TM) region. DI has an N-linked glycan (N154).

**Figure 3 vaccines-10-00834-f003:**
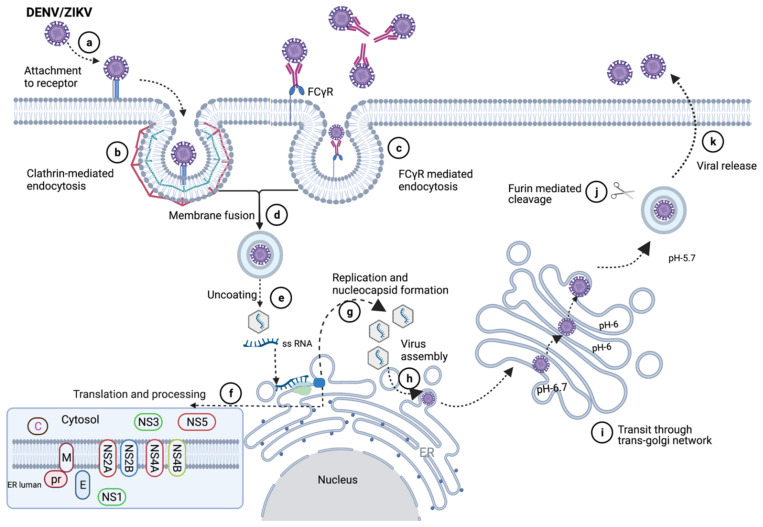
DENV/ZIKA virus entry and release pathway [28]. The virus attaches with the cellular receptors (a) it undergoes Clathrin-mediated (b) or FCγ mediated endocytosis (c), the virus fuses with the endosomal membrane (d), virus genetic material releases in the cytoplasm (e), translates to protein (f), replication and nucleocapsid formation occurs (g), virus assembles in ER (h), immature virus moves through the trans-golgi network (i), cleaved via Golgi protease (j), and the virus releases outside of the cell (k).

**Figure 5 vaccines-10-00834-f005:**
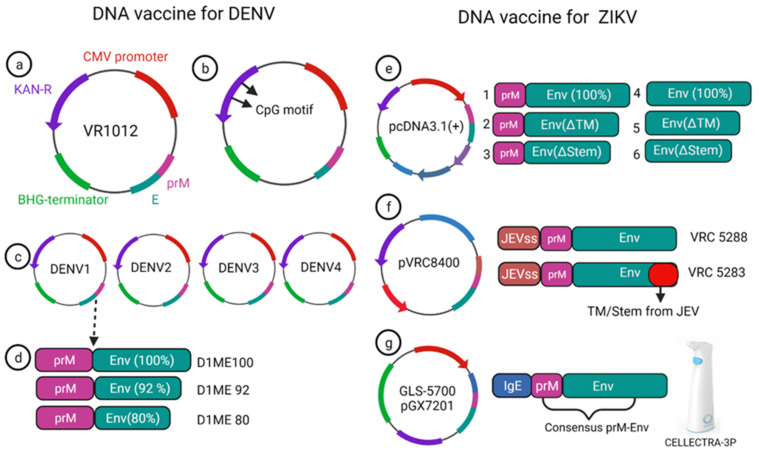
DNA vaccine construct designs for DENV and ZIKV. DNA vaccine for DENV(**a**–**d**); VR1012 construct is used to clone the prM and E gene under CMV promoter (**a**), CpG motif incorporation to the plasmid (**b**), tetravalent vaccine construction for all four dengue (**c**), DENV1 vaccine construction using prM and full length or a truncated version of E protein (**d**). DNA vaccine for ZIKV (**e**–**g**); prM-Env and other deletion mutant (**e**), JEV signal sequence containing prM and Env vaccine VRC 5288 have a WT TM/stem sequence, whereas VRC 5283 have a JEV TM/Stem (**f**), GLS-5700 vaccine containing consensus prM-Env with IgE sequences which delivered using a CELLECTRA-3P device (**g**).

**Figure 6 vaccines-10-00834-f006:**
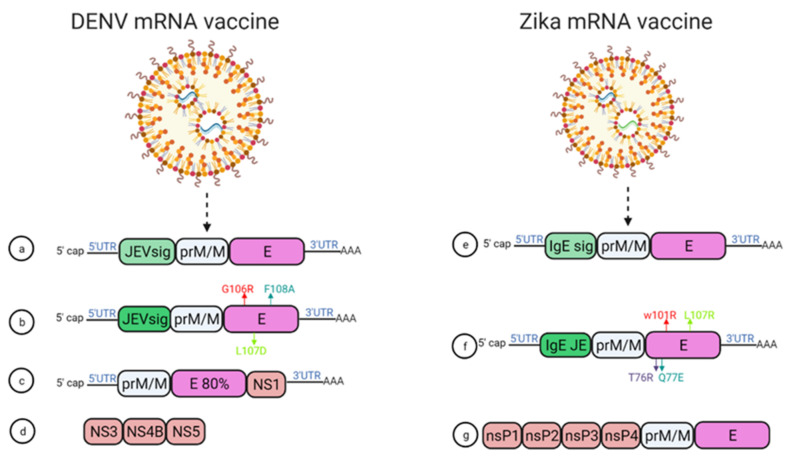
mRNA vaccine construction for DENV and ZIKV. DENV mRNA vaccine (**a**–**d**), Zika mRNA vaccine (**e**–**g**).

**Table 1 vaccines-10-00834-t001:** Current DNA/mRNA vaccine in clinical trial for ZIKA (As of 20 February 2022, from ClinicalTrials.gov).

Platform	Vaccine Name	Antigen	NCT Number	Clinical Trial Stage
DENV DNA vaccine	D1ME100	prM/E DENV1	NCT00290147	Phase I
	TVDV	prM/E DENV1-4	NCT01502358	Phase I
ZIKV DNA vaccine	VRC5283	prM-E	NCT02996461	Phase I
	VRC5283	prM-E	NCT03110770	Phase II
	VRC5288	prM-E	NCT02840487	Phase 1
	GLS-5700	prM-E	NCT02809443	Phase I
ZIKV mRNA vaccine	mRNA-1325	prM-E	NCT03014089	Phase I
	mRNA-1893	prM-E	NCT04064905	Phase I

## Data Availability

Not applicable.

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
