# Peer review of "Nucleic Acid Vaccine Platform for DENGUE and ZIKA Flaviviruses"

_vaccines, 2022, doi:10.3390/vaccines10060834_

Round 1

Reviewer 1 Report

Mourosi et al. reviewed the nucleic acid vaccine platform and the progress of the DNA/RNA vaccine candidates for DENV and ZIKA viruses. Overall, the manuscript is well-organized and useful review for the reader. However, there are many typos.

Line 12, It is recommended to change “Dengue and Zika” to “Dengue virus and Zika virus”. The same applies below

Line 48, Dengue to Dengue virus

Line 68, NS41 to NS4A

Line 71, PrM to prM

Line 72, NS41 to NS4A

Line 101, (HSP70), to (HSP70).

Line 181, invitro to in vitro.

Line 253, “8and to 8 and

Line 272, Fig. 6c to Fig. 6d?

Line 276, Fig. 6d to be Fig.6c?

Line 163, Fig.4 There is no “(c)” in the legend

Line 232, The reviewer did not recognize “the prM-E vaccine” construct. Is this represent 1 (Env 100%) of Fig.5e? Please make this point clear.

Line 292, What is a “JEV promoter”?.  Is this a typo?  signal sequence?

Line 309, Please describe which animal was used in this experiment.

Line 190 Blair et al., 2006   Is a year required? The same applies below

Line 238 Dowd et al., 2016  

Author Response

Mourosi et al. reviewed the nucleic acid vaccine platform and the progress of the DNA/RNA vaccine candidates for DENV and ZIKA viruses. Overall, the manuscript is well-organized and useful review for the reader. However, there are many typos.

We have corrected all the typos in the manuscript. We have also added few lines in introduction about Dengviaxia’s and improved the conclusion.

Line 12, It is recommended to change “Dengue and Zika” to “Dengue virus and Zika virus”. The same applies below

Corrected

Line 48, Dengue to Dengue virus

Corrected

Line 68, NS41 to NS4A

Corrected

Line 71, PrM to prM

Corrected

Line 72, NS41 to NS4A

Corrected

Line 101, (HSP70), to (HSP70).

Corrected

Line 181, invitro to in vitro.

Corrected

Line 253, “8and to 8 and

Corrected

Line 272, Fig. 6c to Fig. 6d?

Corrected

Line 276, Fig. 6d to be Fig.6c?

Corrected

Line 163, Fig.4 There is no “(c)” in the legend

Corrected

 Line 232, The reviewer did not recognize “the prM-E vaccine” construct. Is this represent 1 (Env 100%) of Fig.5e? Please make this point clear.

Author added some sentences (line 354-358)  for clarification.

Line 292, What is a “JEV promoter”?.  Is this a typo?  Signal sequence?

It was a typo. It is corrected.

Line 309, Please describe which animal was used in this experiment.

Information has been added.

Line 190 Blair et al., 2006   Is a year required? The same applies below

Corrected

Line 238 Dowd et al., 2016  

Corrected

Reviewer 2 Report

In this manuscrit Mourosi et al, present a review on DNA and mRNA vaccines candidate for DENGUE and ZIKA viruses. The manuscript is well written and presents a concise review of the literature on the two viruses (their biological cycle, the principle of DNA and RNA vaccines and then presents the different strategies adopted to develop RNA and DNA candidate vaccines and their passage into clinical trial). Changes need to be made to enable this manuscript to be published :

Line 52 : a secondary infection by another serotype may not necessarily result in severe forms for DENV. «can leads to… » is fine to my point.

In figure 2 : the bracket position is confuse with d

The authors focus mainly on prME for the two viruses as candidates. NS1 is also important for vaccines candidate even some debate around this point (For exemple : Costa et al, 2007, Grubor-bauk

 et al, 2019).  Either the authors argue for this choice, or they include a more descriptive section on NS1 (soluble form and physiological effects) and then present RNA and DNA vaccine candidates for this target.

Author Response

In this manuscript Mourosi et al, present a review on DNA and mRNA vaccines candidate for DENGUE and ZIKA viruses. The manuscript is well written and presents a concise review of the literature on the two viruses (their biological cycle, the principle of DNA and RNA vaccines and then presents the different strategies adopted to develop RNA and DNA candidate vaccines and their passage into clinical trial). Changes need to be made to enable this manuscript to be published :

Line 52 : a secondary infection by another serotype may not necessarily result in severe forms for DENV. «can leads to… » is fine to my point.

Corrected

In figure 2 : the bracket position is confuse with d

The bracket is changed with two merging arrows. Other changes have been made in the figure to make it more clear.

The authors focus mainly on prME for the two viruses as candidates. NS1 is also important for vaccines candidate even some debate around this point (For exemple : Costa et al, 2007, Grubor-bauk et al, 2019).  Either the authors argue for this choice, or they include a more descriptive section on NS1 (soluble form and physiological effects) and then present RNA and DNA vaccine candidates for this target.

Few lines have been added about NS1 as vaccine candidate for DENV and ZIKV. As we discussed mainly current clinical trial vaccine constructs, NS1 part was missing earlier, but we agree with the reviewer, NS1 might be potential target despite some controversies.
